# Nanoparticle Mediated Plant Tolerance to Heavy Metal Stress: What We Know?

Mohammad Faizan [1,*,†], Pravej Alam [2,†], Vishnu D. Rajput [3], Ahmad Faraz [4], Shadma Afzal [5], S. Maqbool Ahmed [1], Fang-Yuan Yu [6], Tatiana Minkina [3] and Shamsul Hayat [7]

1 Botany Section, School of Sciences, Maulana Azad National Urdu University, Hyderabad 500032, India
2 Department of Biology, College of Science and Humanities, Prince Sattam Bin Abdulaziz University, Alkharj 11942, Saudi Arabia
3 Academy of Biology and Biotechnology, Southern Federal University, 344006 Rostov on Don, Russia
4 School of Life Sciences, Glocal University, Saharanpur 247121, India
5 Department of Biotechnology, Motilal Nehru National Institute of Technology Allahabad, Prayagraj 211004, India
6 Collaborative Innovation Center of Sustainable Forestry in Southern China, College of Forest Science, Nanjing Forestry University, Nanjing 210037, China
7 Plant Physiology Lab, Department of Botany, Aligarh Muslim University, Aligarh 202002, India
* Correspondence: faizanetawah8@gmail.com; Tel.: +91-7417659443
† These authors contributed equally to this work.

**Abstract:** Nanoparticles (NPs) are playing an important role in addressing various environmental constraints by giving ingenious and successful resolutions. Heavy metal (HM) stress has gained significant importance in the last few years because of its speedy incorporation into agricultural sectors. Due to exclusive physiochemical properties, NPs can be effectively applied for stress mitigation strategies. NPs are highly effective over bulk scale parts owing to the control of the enhanced surface area and the possibility for specific properties to enhance nutrient uptake. In the present review, we explore the use of NPs as an environmentally sound practice to enhance plant growth when exposed to abiotic stress, particularly HM stress. Furthermore, we display an extensive summary of recent progress concerning the role of NPs in HM stress tolerance. This review paper will also be useful for comprehending phytoremediation of contaminated soils and indicates the prospective research required for the cooperative submission of NPs in the soil for sustainable agriculture.

**Keywords:** heavy metal; nutrient uptake; sustainable agriculture; soil health

## 1. Introduction

Important technological progressions and modernizations have been made in current years in the field of crop growing to tackle the difficulties of sustainable manufacture and food safety [1]. Continuous agricultural innovations, such as the use of nanotechnology, are important to feed the rapidly growing world population. The term "Nanotechnology" was coined by Eric Drexler [2], and involves dealing with materials whose structures exhibit appreciably unique and enhanced chemical, physical, and biological possessions as a result of their particle size in the nanoscale [3]. Nanotechnology has great effects on the commercial implementation of nano minerals in the fields of information technology, engineering, medicine, food, and pigments, with pharmaceutical, biological, and electrical applications [4].

It is broadly understood that agricultural output needs to be increased to feed a global population which is carefully expected to be 2 billion in the next 30 years. The NPs are micro-elements that are extremely fine in nature with sizes ranging from 1–100 nm in at least two of their dimensions [5]. Garcia-Lopez et al. [6] reported that NPs play significant roles in plant growth, yield, and quality. They have existed in nature since the beginning of

the Earth's history in the form of ashes from volcanoes and woods fires, oxides of metals, and canals and oceans [7]. NPs possess unique catalytic properties [8], and remarkable progress has been noted during the past few years regarding their special features such as the high surface area to volume ratio and reactivity in comparison with bulk sized materials [9]. The effects of NPs on plants growth, photosynthesis, and hormonal content are described below in Table 1.

Potential NPs innovations have been employed in agriculture biotechnology, food safety, water and sewage treatment, biosensing, clinical diagnosis, and therapy application [10]. The input of NPs to enhance plant growth and productivity with the objective to increase quality and overall production of crops is presently being researched all over the world. NPs are of different types such as inorganic NPs, carbon-based NPs, dendrimers, and composites [11]. Carbon-based NPs include fullerenes and carbon nanotubes (single-walled and multi-walled), whereas inorganic NPs are broadly divided into metals (Au (gold), Ag (silver), and Al (aluminum), and metal oxides ($ZnO$, $TiO_2$, $Fe_2O_3$, $NiO$, $CoO$, $CeO_2$, etc.). Nanosized polymer networks built from branched units are considered dendrimers, while composite NPs are made by combining one type of NPs with another or with larger materials. NPs have been already successfully applied in agriculture and in environmental applications. Several studies have revealed that NPs could improve plant seed germination, plant photosynthesis, and resistance to oxidative stress, as well as crop yield and quality [12–20].

**Table 1.** Impacts of nanoparticles on plants.

| Nanoparticles | Test Crop | Effect of Nanoparticles | Reference |
|---|---|---|---|
| ZnO | *Cyamopsis tetragonoloba* (Cluster bean) | Significant increase in chlorophyll content, leaf protein, and alkaline phosphate | [12] |
| | *Vigna radiata* | Enhanced germination and growth | [13] |
| | *Arachis hypogaea* | Increased seed germination rate | [14] |
| | *Cicer arientum* | Enhanced weight of the plant | [15] |
| | *Glycine max* | Increased root growth and development | [16] |
| CeO₂ | *Glycine max* | Enhancement in plant growth and development | [17] |
| | *Triticum aestivum* | Enhanced shoot growth, biomass, grain yield | [18] |
| | *Zea mays* | Increased stem and root growth | [16] |
| | *Sorghum* | Enhanced assimilation rates of carbon in leaf and seed yield | [19] |
| | *Allium cepa* | Improved growth, yield, and nutrient content | [20] |
| Ag | *Lolium multifolium* | Enhanced plant growth and development | [21] |
| | *Eruca sativa* | Increased root length | [21] |
| | *Zea mays* | Increased root length | [22] |
| | *Oryza sativa* | Enhanced root length | [23] |
| SiO₂ | *Lycopersicum esculentum* | Increased seed germination | [24] |
| Al₂O₃ | *Triticum aestivum* | Enhanced root growth | [25] |
| | *Glycine max* | Improved survival and growth | [26] |
| | *Zea mays* | Root length augmented | [27] |
| | *Glycine max* | Significant enhancement in root length | [28] |
| | *Raphanus sativus* | Improved root growth | [27] |

**Table 1.** *Cont.*

| Nanoparticles | Test Crop | Effect of Nanoparticles | Reference |
|---|---|---|---|
| TiO$_2$ | *Triticum aestivum* | Increased root length | [29] |
| | *Rosa* | Enhanced plant resistance to infection of fungi by changing endogenous hormonal content | [30] |
| | *Spinacia oleracea* | Enhanced growth | [31] |
| | *Lemna minor* | Improved the activities of CAT, SOD, and POX by eliminating accumulated ROS in plant | [32] |
| | *Cicer arietinum* | Altered redox status | [33] |
| | *Spinacia oleracea* | Dry weight and chlorophyll level increased | [34] |
| | *Spinacia oleracea* | Increased fresh and dry biomass | [31] |
| | *Spinacia oleracea* | Improved light absorbance and carbon dioxide assimilation | [35] |
| Fe/Fe$_2$O$_3$ | *Citrullus lanatus* | Enhanced the activity of root, POX, SOD | [36] |
| | *Triticum aestivum* | Enhanced seed germination rate and plant growth | [37] |
| | *Triticum aestivum* | Increased shoot and root biomass | [26] |

CAT: Catalase; POX: Peroxidase; SOD: Superoxide dismutase.

Heavy metals are included as one of the main abiotic stresses which reduces plant development and productivity all over the world (Figure 1) [38]. At present, extensive anthropogenic activities relay negative effects on plant development by accumulating HMs, and an increase in their concentration is immensely dangerous because HMs are toxic and mostly non-degradable in nature [39,40]. In soil, HMs get accumulated after leaching or being released after the oxidation process, making them easier to uptake by plants and ultimately affecting public health via the food chain [41].

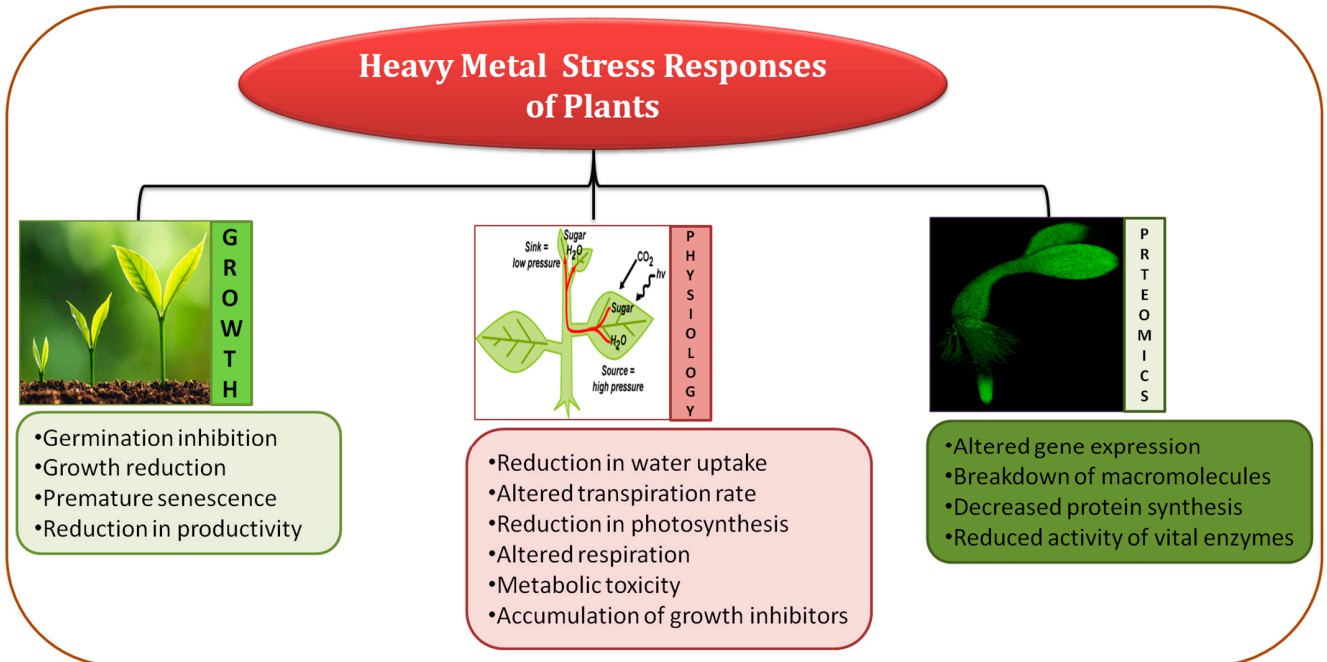

**Figure 1.** Diagrammatic representation of heavy metal effects on plant growth and development.

Accumulation of metals instigates direct or indirect harmful effects causing morpho-physiological abnormalities (decreased photosynthesis, plant growth, biomass, and yield), giving rise to the formation of ROS resulting in redox homeostasis disruption, damaging plasma membrane of cells, proteins, lipids, and nucleic acids, and causing disorder in

metabolic activities that ultimately lead to reduced agricultural productivity [40,42,43]. Therefore, increasing plant tolerance against HM stress is a significant goal to diminish worsening food construction and to meet the order for increased food delivery for the increasing global population [44].

In order to reduce HM toxicity in plants, exogenous application of NPs has emerged as a cost-effective strategy [45], and various studies have confirmed their positive impacts on plant growth and development (Table 1). For example, in a study performed by Wang et al. [45], $Fe_3O_4$ NPs stimulated root growth of *Cucumis sativus*, *Solanum lycopersicum* L., *Lactuca sativa*, and *Daucus carota* in the presence of Cd. According to a recent study by Faizan et al. [46], the application of ZnO NPs and SA to rice plants reduced the toxicity of As. As-applied plants showed reduced plant growth, gas exchange indices, maximum quantum yield, and a reduction in protein content. As-stressed rice plants receive foliar fertigation (ZnO NPs and/or SA,) which reduces the oxidative stress as seen by lower synthesis the levels of ROS. Under the influence of ZnO NPs and salicylic acid (SA), the enzymatic activities of CAT, SOD, and POX, as well as the contents of proline and total soluble proteins, all improved, all of which play a crucial role to control various transcriptional pathways involved to tolerate oxidative stress.

NPs enhance resistance of plants against several toxic elements i.e., Cr, Cd, Fe, Al, and Mn [47–49]. NP application has also been known to reduce the uptake and translocation of Cd [50], Mn [51], and Pb [52] in plant tissues. Moreover, NPs play an important role to protect plants against environmental stresses by boosting antioxidant activity, accumulation of osmolytes, and synthesis of free amino acids and nutrient enhancements (Figure 2) [53]. Doubtlessly, a huge void of information regarding the valuable functions of NPs in the regulation of metal stress is yet to be filled. This review highlights the mechanism of plant growth inhibition induced by HMs as well as the role of NPs in HM stress tolerance.

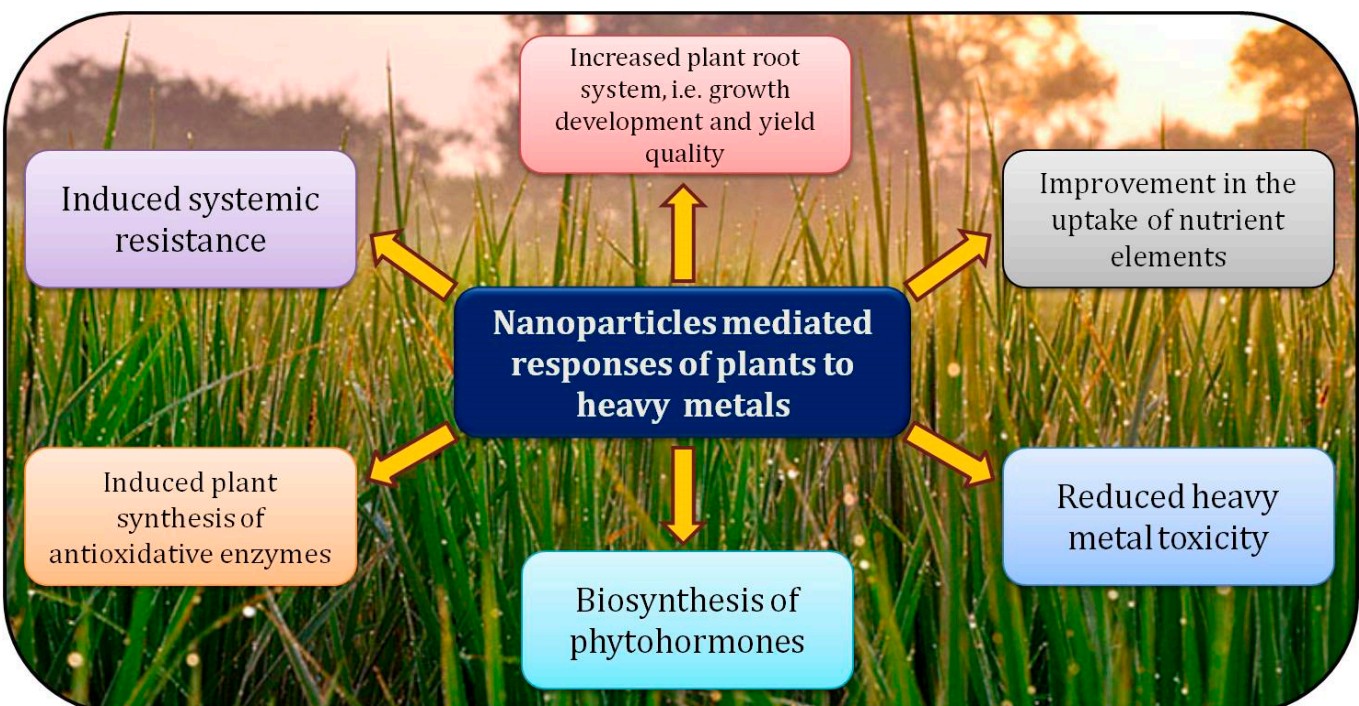

**Figure 2.** The multiple functions of nanoparticles alleviate heavy metal stress in plants.

## 2. Mechanism of Plant Growth Inhibition Induced by Heavy Metals

Among others, some HMs, i.e., Cd, Pb, Cr, As, Fe, Mn, and Al have been deemed the most effective metals. Below is a brief description of the impact of these HMs on plant growth, development, and yield.

### 2.1. Cadmium (Cd)

Being one of the most notorious environmental pollutants, Cd causes significant adverse impacts on all organisms, affecting the health of plants and humans primarily via the food-chain through edible crops grown in Cd-polluted soil [54,55]. Geogenic (weathering of parent rocks and volcanic activity) and anthropogenic activities (irrigation with sewage, use of phosphate fertilizers and pesticides, manure application, fossil fuel combustion, mining, and smelting, as well as atmospheric precipitation) are the major Cd contributors in cultivable soils and fresh water reservoirs [56,57]. Owing to its potentially threatening demeanor (being readily absorbed, translocated, and accumulated in living cells, having a longer half-life and a non-biodegradable nature), it has escalated concerns among agriculturists and environmentalists [58].

Plants exposed to Cd toxicity have been reported to exhibit reduced biomass, photosynthesis, and productivity attributed to slower functioning at a morphological, physiological, and molecular level [59,60]. Saifullah et al. (2014) [61] elaborated that stomatal conductance and enzyme activities of plants can reduce post-Cd application. Moreover, the presence of Cd could enhance ROS generation and lead oxidative stress through enhanced $H_2O_2$ content, posing a threat to membrane integrity due to lipid peroxidation and overall damage to cellular proteins, nucleic acids, and lipids, which in extreme cases, could finally result in cell death [62].

### 2.2. Lead (Pb)

Heavy metal pollutants such as Pb can enter the soil via several ways and induce a broad range of toxicity to living organisms [63]. This metal remains stable in soil for a long period of time, is difficult to dissociate, and with the help of food chain, it eventually accumulates in the cells of living organisms including human beings [64]. Pb impairs cell division, growth, and performance of plants as well as chlorophyll contents [65].

In particular, seed germination rate is strongly affected by Pb and may result from its interference with a plant's protease and amylase enzymes [66]. At low concentrations, Pb potentially causes loss of cristae, mitochondrial swelling, vacuolization of endoplasmic reticulum, and bent, short, and stubby roots [67]. Photosynthetic machinery and nutrient allocation may also deteriorate after lead exposure followed by growth retardation and plant death [68].

### 2.3. Chromium (Cr)

Cr was discovered by a French chemist Vauquelin and stands as the seventh most abundant element on the Earth's crust [69]. The contamination of soil, sediments, and groundwater by this toxic metal may cause deleterious effects on plants and animals [70]. Cr is a non-essential element for plants; hence, they do not have any special mechanism for its uptake but due to its high solubility and oxidizing potential it becomes detrimental to their growth and development [71]. In nature, Cr is present in different oxidation states; among them trivalent ($Cr^{3+}$) and hexavalent ($Cr^{6+}$) are the most stable forms [72]. $Cr^{6+}$ is considered to be more damaging than $Cr^{3+}$ because of its association with $O_2$ as chromate or dichromate; conversely, $Cr^{3+}$ is associated with organic matter in soil [73].

Cr reduces seed germination by having effects on the activities of amylase and on the consequent sugars transport to the embryo axes; however, the activities of protease enzymes increase with its application, which may contribute to the decrease in seed germination, particularly of Cr-exposed seeds [74]. Barton et al. [75] demonstrated that the addition of Cr significantly inhibited the shoot growth in lucuma cultures. The leaf growth and size of spinach plants grown in 60 mg Cr $kg^{-1}$ soil showed a marked decrease in growth rate and caused burning of leaf tips and/or margins [76]. In the study of Vajpayee et al. [77] on *Vallisneria spiralis*, Cr application in the nutrient medium considerably reduced biomass production and dry weight of the plant. The decrease in the dry matter is an indicator of Cr toxicity which decreases the size of the peripheral part of the *Portulaca oleracea* [78].

*2.4. Iron (Fe)*

Fe is an important element for plants [79,80]. It plays a vital role as a co-factor for various enzymes that are involved in respiration, photosynthesis, chloroplast synthesis, nucleic acid synthesis, enzyme catalyzed processes, metal homeostasis, and maintaining the structural and functional integrity of proteins and chlorophyll [81,82]. Moreover, Fe assists in energy production within the plants. However, a growth medium having low pH or an excessive application of Fe can induce toxicity in most plant species [83]. Plants suffering from Fe deficiency may exhibit symptoms of interveinal chlorosis and stunted growth. Discoloration of the leaf is due to the plant crating enzymes to control free radicals that are present in elevated Fe condition. The phytotoxicity of Fe could be of two types, i.e., direct and indirect [84], where absorption of the metal and accumulation in the plants represents the direct type of Fe toxicity, while Fe plaque formation on the root surface (which affects nutrient absorption) falls under the second category [85].

*2.5. Manganese (Mn)*

Mn is an important mineral nutrient for plants, and plays a pivotal role in several physio-biochemical processes. In crop plants, excessive Mn is disadvantageous and may significantly reduce fresh and dry biomass and photosynthetic activities, as well as cause biochemical disorders [86]. It is readily transported from root to shoot via transpiration stream; however, it is not readily remobilized thorough phloem to the other tissues or organs once it reaches the leaves [87] and its toxicity is a fairly general problem over other micronutrients. Elevated contents of Mn may result in necrotic brown patches on plants leaves and petioles, as well as on stems [88].

**3. Role of Nanoparticles on Heavy Metal Stress Tolerance**

The application of NPs not only shows positive impacts on plants but also is effectual in eliminate metals from polluted sites using possibilities such as a phytoremediation approach [88]. Different types of NPs are developed based on the application and usage, such as those containing inorganic nonmetallic NPs, carbon-based NPs, metallic NPs, and organic polymeric materials. Plant-based methods known as phytoremediation are used for clearing or removing a number of soil impurities; they are interesting procedures that revolve around a plant–nanoparticles interaction to promote plant growth, increase phyto-availability of pollutants, and eliminate toxic pollutants from contaminated soil [89]. Phytoextraction is one of the most important remediation procedures for the Pb elimination [90], and the input of NPs with plants has been noted to be effective for enhancing Pb phytoextraction. A study showed that the co-application of *Lolium* (ryegrass) and nanohydroxyapatite eliminated 30% of Pb from the soil after one month and 44.39% after 3 months [91].

HM pollution is very hazardous for living organisms, which raises concern all over the world [92]. These metals exist in our environment in small quantities due to natural processes like volcanic eruption and rock weathering. In the last few decades, concentrations of these HMs have increased in the environment due to anthropogenic activities like mining, fossil fuel combustion, and poor drainage systems of factories [93]. Soil becomes the ultimate reservoir for all the released HMs, including Cd, Pb, Zn, and Cu, which are easily taken up by plants through their roots when grown in heavy metal-laced agricultural lands. When these HMs enter the plant tissues, they affect the plant life cycle and cause stunted growth and lower yields [94]. Pb and Cd metal ions are released into farming lands, air, and water, causing important environmental issues because of their deleterious effect on human health and the ecosystem [95]. Several investigations indicate that some crops are hyper accumulators of HMs [17,63]; these plant species have adapted to endure the toxicity while accumulating heavy metals in their tissues. These HM accumulators are more harmful for human beings because they sneak the HMs into our food chain. It has been reported that more than 80% of HMs are ingested via food crops [96]. A study also found that during 1998–2001, native populations consumed more than 35% of Cd from rice [97].

Some HMs, like Zn and Mn, are beneficial for plants and animals at lower concentrations; the situation reverses and they become toxic to plants once their concentration increases above a threshold value, and they contaminate the soil and water [98,99].

To remove these metals from contaminated soils, different techniques such as electronic remediation, biosorption, bioleaching, and bioremediation are employed [45]. Application of silicon dioxide improved the Cd, Cu, and Mn stress tolerance potential of *Artemisia pygmaea* by augmenting biomass accumulation and increasing the activities of different biocatalysts in the plant. In the last few years, nanotechnology has become a well-established field in the agricultural realm. Numerous pieces of evidence have shown the ability of NPs to improve plant growth, development, and yield, as presented in Table 1 [100–102]. Nanotechnology could be proposed as an important tool for environmental cleansing. Studies have shown that NPs have special properties which can be utilized to reduce the harmful effects of natural HMs resources (Table 2).

**Table 2.** Impacts of nanoparticles to alleviating abiotic stresses in plants.

| Nanoparticles | Test Crop | Effect of Nanoparticles | Reference |
|---|---|---|---|
| TiO$_2$ | *Triticum aestivum* L. | Altered growth, yield, and starch level in drought stress | [103] |
| | *Oryza sativa* L | Altered Cd migration in the soil–rice system in Cd toxicity | [104] |
| | *Oryza sativa* L. | NPs decreased the Cd uptake in Cd toxicity | [105] |
| | *Linum usitatissimum* L. | Altered length, biomass, yield, and carotenoid level in drought stress | [106] |
| | *Oryza sativa* L. | Mitigated Cd toxicity by improving antioxidant enzymes activity | [94] |
| | *Ocimum basilicum* L. | Reduced toxicity caused by drought stress | [107] |
| SiO$_2$ | *Lycopersicum esculentum* L. | Reduced salinity stress and enhanced seed germination potential, root length, and dry weight | [24] |
| | *Solanum lycopersicum* L. | Altered weight and photosynthetic attributes under salinity stress | [108] |
| | *Ocimum basilicum* L. | Ameliorated salt stress and increased fresh and dry weight, chlorophyll content, and proline content | [109] |
| | *Lens culinaris Medik* | Altered seed germination rate and plant growth under salinity stress | [110] |
| | *Cucurbita pepo* L. | Increased activities of CAT, POX, and SOD | [111] |
| ZnO | *Moringa peregrina* | Decreased salt stress and reduction in Na$^+$ and Cl$^-$ contents | [112] |
| | *Helianthus annuus* L. | Reduced salinity stress, increasing net CO$_2$ assimilation rate | [113] |
| | *Gossypium barbadense* L. | Increased growth, yield, and mineral contents in salinity stress | [114] |
| | *Helianthus annuus* L. | Reduced salinity stress, increased plant growth, net CO$_2$ assimilation | [113] |
| | *Glycine max* | Mitigate drought stress and increasing seed germination rate | [115] |
| | *Sorghum bicolor* | Enhanced drought tolerance, accelerated plant development and promoted yield | [116] |

Furthermore, there is less literature on the potential role of NPs in mitigating the phytotoxic effects caused by HMs and removal of HMs from contaminated soil and waters but what exists is very convincing [117–119]. Due to an increase in the concentration of HMs in soil and water worldwide because of human activities, there is an urgent need to



solve this problem so as to contain their harmful effects on plants and animals. Using the NPs to delimit HMs toxicity is one link among several. Nanotechnology in agriculture is used to improve plant growth and development due to the unusual characteristics of NPs as many experiments during the last few years have supported the notion that the toxic effects of HMs in plants could be checked by NPs application. Rizwan et al. (2019) [94] reported that ZnO NPs, either individually or in combination with biochar, effectively alleviate the Cd toxicity in maize plants when given as a foliar spray.

Although, NPs can be low-cost approach to reduce HMs toxicity in plants, their role in ameliorating oxidative stress and root-growth inhibition caused by HMs has been seldom studied [45,63]. Results from trials conducted to find the effects of $Fe_3O_4$ in Cd, Pb, Zn, and Cu toxicity in wheat seedlings show that $Fe_3O_4$ effectively alleviated the inhibition caused by these HMs; the goal was achieved mainly by protecting the plants from oxidative stress with increased antioxidant enzyme activity [92]. Moreover, Cui et al. (2017) [120] reported similar results where Si NPs reduced the Cd toxicity in rice. Their findings also suggest that Si NPs have the ability to protect against cell damage caused by Cd. Similarly, Wang et al. (2015) [121] revealed that a Si NP hydrosol can relieve toxicity generated by Cd in rice by diminishing its uptake, lowering the malondialdehyde (MDA) level, increasing the assimilation of certain mineral components, and triggering antioxidant activity. Si NP hydrosol application was presented as an economical and environmentally friendly approach to minimize the accumulation of Cd in rice by Liu et al. [122]. Root growth inhibition was also reduced by $Fe_3O_4$ NPs in four plants (tomato, cucumber, lettuce, and carrot) under Cd stress [92]. Protection against oxidative stress caused by Cd and Pb in *Leucaena leucocephala* was reported with ZnO NPs, mollifying the toxic effects of these heavy metals [123].

Foliar-applied ZnO NPs on tomato plants exposed to Cd metals also mitigated their deleterious effects and improved plant growth by increasing the antioxidant enzymes and nitrate reductase activity [124]. Very interesting results were reported by Sing and Lee (2016) [88] with $TiO_2$ NPs. They found that soil contaminated with Cd decreased the plant growth, biomass, and protein content in soybean plants, while the application of $TiO_2$ NPs not only improved the photosynthetic rate and growth but also increased the uptake of Cd from soil to plant, thus minimizing their toxic effects. Moreover, Li and Huang [125] and Faraji and Sepehri (2018) [50] reported that NPs alleviate the HMs toxicity in *Brassica chinensis* and wheat, respectively. Nano hydroxyapatite increased the chlorophyll level and vitamin C and decreased the MDA content in *Brassica chinensis* under Cd stress. In conclusion, we find that in most cases NPs protect the plants from oxidative stress and cell disintegration when exposed to HMs, increasing the antioxidant enzyme activity and reducing HM uptake in plants.

Overall, these studies show that NPs have a potential to mitigate the HMs toxicity in plants and could improve plant growth, biomass, and yields. This necessitates further research in order to find the underlying mechanism of NPs in the combat against HM toxicity. Although most results talk about the alleviation of Cd toxicity by NPs, extensive research is still indispensable to explore the effects with different HMs and different crops. It was also reported that rGO (reduced grapheme oxide) hybridized with magnetic and/or elemental silver nanoparticles (rGO/magnetite, rGO/magnetite/silver and rGO/silver) is a potential adsorbent for noxious heavy metals such as Cd, Ni, Zn, Pb, Cu, and Co. These nanohybrids (NHs) showed more adsorption efficiency than the sorbents already reported, including resins, activated biochar, and hydrated nano-sized Zn oxide particles [126].

## 4. Conclusions

Crop production globally has undergone several challenges in terms of climate and stresses. Heavy metal contamination and remediation approaches have received significant attention worldwide due to the non-biodegradable nature of many HMs, whose persistence in the soil for a long period of time can be dangerous to soil biota as well as human health. Many strategies have been successfully applied to enhance the tolerance of plants to grow

in metal-polluted soils, whether by tolerating or accumulating excessive metal in plant tissues. To overcome the hazardous effects of HMs, nanotechnology has come up as a key component for sustainable development. The input of NPs in plants to tolerate HM toxicity and for remediation of these toxic elements is the best and most economic approach recently developed. Due to their properties of being able to penetrate plants over a large surface area, more effective adsorption, and targeted delivery, they can be responsible for helping regulate photosynthesis and detoxifying reactive oxygen species, thereby enhancing seed germination, growth, and yield of crops. NPs also helps plants to grow well by modulation of mobile or immobile forms of HMs. The positive results exhibited by the application of NPs, specifically increasing HM tolerance and promoting the growth ability of plants, indicate that their application in remediating metal-polluted soils might have noteworthy potential in the near future.

**Author Contributions:** Conceptualization, S.H., T.M. and F.-Y.Y.; methodology, M.F.; writing—original draft preparation, P.A., V.D.R., A.F. and S.A.; writing—review and editing, S.M.A.; visualization, P.A.; supervision, S.H. All authors have read and agreed to the published version of the manuscript.

**Funding:** This research received no external funding.

**Institutional Review Board Statement:** Not applicable.

**Informed Consent Statement:** Not applicable.

**Data Availability Statement:** Not applicable.

**Acknowledgments:** V.D.R. and T.M. acknowledge the support of the Russian Science Foundation, project no. 21-77-20089 at the Southern Federal University.

**Conflicts of Interest:** The authors declare no conflict of interest.

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
