# Peer review of "Nanoparticle Mediated Plant Tolerance to Heavy Metal Stress: What We Know?"

_sustainability, doi:10.3390/su15021446_

Round 1
Reviewer 1 Report
Comments attached.

Author Response
Author responses to reviewer (1) comments
(Sustainability-2066081 R1)
To Reviewer #1
Remarks: Abstract: The abstract could be more concise
Response: Corrected as per your suggestion.
Remarks: At the end of introduction, objective of this study should clearly describe.
Response: Corrected as per your suggestion.
Remarks: Several mistakes in text including, Line 65. Correct it.
Response: Corrected as per your suggestion.
Remarks: Conclusion should be improved.
Response: Corrected as per your suggestion.
Remarks: Grammar should be improved.
Response: Checked and corrected.
Remarks: English language is not very good, check it.
Response: Corrected as per your suggestion.
Remarks: Spacing problem occurs within the text, correct it.
Response: Corrected as per your suggestion.
Remarks: Line 85-87 not clear, explain again.
Response: Corrected as per your suggestion.
Remarks: Line 100-103 not clear, rewrite.
Response: Corrected as per your suggestion.
Remarks: Within the text some species names written scientifically and some are as common. Follow the same pattern.
Response: Corrected as per your suggestion.
All the suggestions and comments of the reviewer have been accepted by the authors and the manuscript has been corrected accordingly. A thorough internal reviews was also performed in the whole MS for possible improvement, changes highlighted in Track Change Format supplied MS. We hope the response meets the reviewer approval.

Reviewer 2 Report
Manuscript entitled “ Nanoparticle Mediated Plant Tolerance to Heavy Metal Stress: What We Know? - has written well and the objective of the work is unique and novel. I appreciate the authors for their efforts .For better clarity, the authors should provide the details
1.In abstract , add the background and clear objective, outcome of the study
2. Provide the methods or techniques used for this study.
3. Write the challenges and future prospects of the study
Author Response
Author responses to reviewer (2) comments
(Sustainability-2066081 R1)
To Reviewer #2
Remarks: Manuscript entitled “Nanoparticle Mediated Plant Tolerance to Heavy Metal Stress: What We Know? - has written well and the objective of the work is unique and novel. I appreciate the authors for their efforts. For better clarity, the authors should provide the details.
Response: Dear Reviewer: We are grateful for the critical comments and thoughtful suggestions provided by you of our manuscript " Nanoparticle Mediated Plant Tolerance to Heavy Metal Stress: What We Know?" Based on the constructive comments and suggestions, we have made careful modifications to the original manuscript and highlighted words.
Remarks: In abstract, add the background and clear objective, outcome of the study.
Response: Corrected as per your suggestion.
Remarks: Provide the methods or techniques used for this study.
Response: It is a review article, no any methods or techniques used in this study.
Remarks: Write the challenges and future prospects of the study.
Response: Mentioned in the conclusion.
All the suggestions and comments of the reviewer have been accepted by the authors and the manuscript has been corrected accordingly. A thorough internal reviews was also performed in the whole MS for possible improvement, changes highlighted in Track Change Format supplied MS. We hope the response meets the reviewer approval.

Reviewer 3 Report
General Comments: The review topic is interesting and valuable to our society to mitigate our increasing agricultural demand. However, the authors need to be well-structured in the manuscript. For that, the manuscript needs to improve by adding more data about – different nanoparticles utilizing in plants’ stress tolerance, mode of their application, mechanism of action and demerits of their application (if any). The abstract portion should be more accurate, informative and precise regarding the work done. Follow the author’s guidelines for mentioning Table no, and Figure no in the text and check the reference section sincerely.
Line No Remarks
61 Remove first bracket
62 These should be written in subscript
66 In Table 1, the common name should be mentioned in bracket and
also mention the scientific name positively. Maintain a common format for all the mentioned plants. Mention the full form below the table for CAT, SOD and POX.
73 up take, should be written as uptake
95 Mention the full form
99 Mention as "such as or i.e."
102 Correct the spelling
103 Write as per Author's guideline (Figure 2)
112, 131 Mention the symbol in bracket
196 Mention the scientific name
223 References are not mentioned in Table 1 [100, 101, 102]
213 A spacebar needed
242,243, 253 Use subscript
249 Use full form and mention abbreviation in bracket
280 Correct the spelling
300 In reference section year should be written in bold and mention doi
as many as possible.
308, 316 "a” and “b" should not be mentioned as it provides the serial number
435 Check it and written as per author's guideline.
473 Check it

Author Response
Author responses to reviewer (3) comments
(Sustainability-2066081 R1)
To Reviewer #3
Remarks: The review topic is interesting and valuable to our society to mitigate our increasing agricultural demand. However, the authors need to be well-structured in the manuscript. For that, the manuscript needs to improve by adding more data about – different nanoparticles utilizing in plants’ stress tolerance, mode of their application, mechanism of action and demerits of their application (if any). The abstract portion should be more accurate, informative and precise regarding the work done. Follow the author’s guidelines for mentioning Table no, and Figure no in the text and check the reference section sincerely.
Response: Dear Reviewer: We are grateful for the critical comments and thoughtful suggestions provided by you of our manuscript " Nanoparticle Mediated Plant Tolerance to Heavy Metal Stress: What We Know?" Based on the constructive comments and suggestions, we have made careful modifications to the original manuscript and highlighted words.
Remarks: Here the report is not only mentioned by Garcia-Lopez et al. [6] in Table 1, there are so many references mentioned in the table. So, all references should be mentioned. Finally reference [6] not mentioned in Table-1. Therefore, I suggest to rewrite.
Response: Corrected as per your suggestion.
Remarks: Remove first bracket
Response: Corrected
Remarks: These should be written in subscript
Response: Corrected
Remarks: In Table 1, the common name should be mentioned in bracket and also mention the scientific name positively. Maintain a common format for all the mentioned plants. Mention the full form below the table for CAT, SOD and POX.
Response: Corrected
Remarks: up take, should be written as uptake
Response: Corrected
Remarks: Mention the full form
Response: Corrected
Remarks: Mention as "such as or i.e."
Response: Corrected
Remarks: Correct the spelling.
Response: Corrected
Remarks: Write as per Author's guideline (Figure 2).
Response: Corrected
Remarks: Mention the symbol in bracket.
Response: Added as per your suggestion.
Remarks: Mention the scientific name.
Response: Added
Remarks: References are not mentioned in Table 1 [100, 101, 102]
Response: Here only given the table name. References 100, 101, 102 are the different from the table 1.
Remarks: A spacebar needed.
Response: Corrected
Remarks: Use subscript.
Response: Corrected
Remarks: Use full form and mention abbreviation in bracket.
Response: Corrected
Remarks: Correct the spelling.
Response: Corrected
Remarks: In reference section year should be written in bold and mention doi as many as possible.
Response: Corrected
Remarks: "a” and “b" should not be mentioned as it provides the serial number
Response: Corrected
Remarks: Check it and written as per author's guideline.
Response: Corrected
Remarks: Check it.
Response: Corrected
All the suggestions and comments of the reviewer have been accepted by the authors and the manuscript has been corrected accordingly. A thorough internal reviews was also performed in the whole MS for possible improvement, changes highlighted in Track Change Format supplied MS. We hope the response meets the reviewer approval.
